# Effect of Nefopam-Based Patient-Controlled Analgesia with and without Fentanyl on Postoperative Pain Intensity in Patients Following Laparoscopic Cholecystectomy: A Prospective, Randomized, Controlled, Double-Blind Non-Inferiority Trial

**DOI:** 10.3390/medicina57040316

**Published:** 2021-03-27

**Authors:** Ki Tae Jung, Keum Young So, Seung Chul Kim, Sang Hun Kim

**Affiliations:** 1Department of Anesthesiology and Pain Medicine, School of Medicine, Chosun University, 309 Pilmun-daero, Dong-gu, Gwangju 61452, Korea; mdmole@chosun.ac.kr (K.T.J.); kyso@chosun.ac.kr (K.Y.S.); 2Department of Anesthesiology and Pain Medicine, Chosun University Hospital, 365 Pilmun-daero, Dong-gu, Gwangju 61453, Korea; flairsizz@empal.com

**Keywords:** intravenous infusion, laparoscopic cholecystectomy, nefopam, opioid analgesics, patient-controlled analgesia, postoperative pain

## Abstract

*Background and Objectives*: We investigated the non-inferiority of patient-controlled analgesia (PCA), using either nefopam alone or combined nefopam-fentanyl for postoperative analgesia in patients undergoing laparoscopic cholecystectomy. *Materials and Methods*: In this prospective, randomized, controlled study, 78 patients were allocated to receive nefopam 240 mg (Group N240) or nefopam 120 mg with fentanyl 600 μg (Group NF), equivalent to fentanyl 1200 μg, with a total PCA volume of 120 mL. Patients were given a loading dose (0.1 mL/kg) from the PCA device along with ramosetron (0.3 mg) and connected to a PCA device with a background infusion rate of 2 mL/h, bolus dose amount set at 2 mL, and lockout interval set at 15 min. Pain scores were obtained using the numeric rating scale (NRS) at 30 min after recovery room (RR) admission, as well as 8 and 24 h postoperatively. The primary outcome was analgesic efficacy evaluated using NRS-rated 8 h postoperatively. Other evaluated outcomes included the incidence rate of bolus demand, rescue analgesic and antiemetic requirements, and postoperative adverse effects. *Results*: NRS scores were not significantly different between the groups throughout the postoperative period (*p* = 0.539). NRS scores of group N240 were not inferior to those of group NF at 30 min after RR admission, or at 8 and 24 h postoperatively (mean difference [95% CI], −0.05 [−0.73 to 0.63], 0.10 [−0.29 to 0.50], and 0.28 [−0.06 to 0.62], respectively). Postoperative adverse effects were not significantly different between the two groups (*p* = 1.000) and other outcomes were also not significantly different between the two groups (*p* ≥ 0.225). *Conclusions*: PCA using nefopam alone has a non-inferior and effective analgesic efficacy and produces a lower incidence of postoperative adverse effects compared to a combination of fentanyl and nefopam after laparoscopic cholecystectomy.

## 1. Introduction

Laparoscopic surgical procedures are preferred over open surgery because of their advantages, such as less postoperative pain, early recovery, and reduced postoperative complications; however, patients often complain of moderate to severe pain after laparoscopic surgery [1] and effective postoperative analgesia is therefore necessary to augment the benefits of laparoscopic surgery and optimize patient satisfaction. Various strategies to address this have been investigated, but the optimal method has not yet been resolved. Recent recommendations focused on multimodal analgesia for a basic analgesic technique, reserving the use of opioids for more severe pain due to opioid-induced postoperative nausea and vomiting (PONV) [2].

Opioid-based patient-controlled analgesia (PCA) is still commonly used for postoperative pain control in many countries and hospitals; however, it is associated with adverse effects, such as nausea and vomiting, respiratory depression, constipation, urinary retention, opioid tolerance, opioid resistance, and opioid-related hyperalgesia [3,4,5]. Postoperative pain intensity typically has a biphasic pattern; it is more intense immediately after surgery and less intense from the day after surgery [6,7]. Therefore, there is a risk of opioid-related adverse effects due to the increased opioid dosing immediately after surgery, and a risk of unnecessary opioid infusion when using a fixed-rate background infusion from the day of surgery [8,9]. 

Postoperative pain management regimens need to minimize or completely avoid the use of opioids whenever possible [3,10]. To minimize opioid use, we have been using PCA regimens using opioid and non-opioid analgesics (especially non-steroidal anti-inflammatory drugs (NSAIDs)) in combinations based on doses of opioid equivalents [11,12]. However, the risk of opioid- and NSAID-related adverse effects remain. 

Nefopam is a centrally acting non-opioid, non-steroidal analgesic that has been used as an alternative to opioids for analgesia in patients with moderate to severe pain [13]. Nefopam, as an adjuvant analgesic for fentanyl-based PCA, has been shown to provide similar postoperative analgesia to ketorolac, a common NSAID used as an adjuvant analgesic with fentanyl-based PCA [14]. Several studies have evaluated the efficacy and safety of nefopam for postoperative analgesia [15,16]. A recent meta-analysis reported that intravenous nefopam infusion was useful in reducing postoperative pain scores, opioid consumption, and opioid-related adverse effects [1]. Another study showed that nefopam alone reduced postoperative opioid consumption, but did not demonstrate a clinically meaningful improvement in postoperative pain [12]. Furthermore, evidence was insufficient to determine whether nefopam reduced postoperative pain effectively, and PCA containing nefopam alone was as effective as that containing a nefopam-fentanyl combination. 

We hypothesized that PCA using nefopam alone could control postoperative laparoscopic pain as effectively as PCA using a nefopam-fentanyl combination. Therefore, we investigated the non-inferiority of PCA, using either nefopam alone or combined nefopam-fentanyl using fentanyl-equivalent doses for postoperative analgesia in patients undergoing laparoscopic cholecystectomy. The primary outcome of this study was whether the non-inferiority margin of the numeric rating scale (NRS) exceeded 1.0 at 8 h postoperatively in the group receiving PCA using nefopam alone (group N240), compared to that using a nefopam-fentanyl combination (group NF). 

## 2. Materials and Methods

### 2.1. Study Design and Ethical Statement

This prospective, randomized, controlled, double-blind non-inferiority study was approved by the Institutional Review Board of Chosun University Hospital (Chosun 2019-05-006) on 6 June 2019, and was prospectively registered with the Clinical Research Information Service (CRIS: https://cris.nih.go.kr/, ref: KCT0002777) accessed on 12 June 2019. It was conducted according to the Declaration of Helsinki of 1964 and all its subsequent revisions.

### 2.2. Selection of Study Population

The subjects included patients aged 20 to 70 years with an American Society of Anesthesiologists (ASA) physical status of I-III who were scheduled to undergo elective laparoscopic cholecystectomy under general anesthesia between 7 June 2020 and 16 December 2020. Written informed consent was obtained from all participants or their legal surrogates after a thorough explanation of the purpose of this study. Participants were instructed to push the “demand” button of the PAINSTOP device (PS-1000, Unimedics Co., Seoul, Republic of Korea) whenever they experienced pain of >4 points on the NRS (0 = no pain, 10 = worst pain imaginable). We excluded patients with renal, hepatic, or thyroid functional abnormalities, neuromuscular disorders, convulsive disorder, moderate to severe respiratory depression, glaucoma, urinary retention, a history of opioid or nefopam medication within 24 h, or a history of opioid- or nefopam-related complications.

### 2.3. Randomization and Masking

Seventy-eight patients were randomly assigned to two groups that received PCA with either a combination of fentanyl 600 μg and nefopam 120 mg (group NF, *n* = 39) or nefopam 240 mg alone (group N240, *n* = 39). Randomization was performed using a computer-generated table of random numbers with a 1:1 allocation ratio. This randomization was performed using an online website (https://www.graphpad.com/quickcalcs/, accessed on 17 May 2019).

The researcher who managed the anesthesia (RA) was responsible for obtaining informed consent from participants, as well as gathering and recording data from the participants and PCA devices. The researcher who managed the PCAs (RP) was responsible for assigning the correct drugs to each PCA device according to the randomization scheme. For blinding, RP recorded the drug assignment on anesthetic charts after the anesthesia was completely finished, and RA finally collated the data from patient medical records, as well as data generated through the trial for at least 24 h postoperatively. Neither RA nor RP participated in the statistical analysis. 

The nurses in the recovery room (RR) or ward recorded data on postoperative pain and postoperative nausea and vomiting (PONV) using the NRS; these nurses were not part of the investigating team and were trained by the hospital to assess pain intensity and PONV using the NRS, visual analogue score (VAS: 0 = no pain, 10 = worst pain imaginable), or Woong–Baker faces pain ratings scale (FPRS: 0 = no pain, 10 = most severe pain). 

### 2.4. Anesthetic Management 

After premedication with intramuscular midazolam (0.05 mg/kg), the patients were transported to an operating room. RA anesthetized the patients using total intravenous anesthesia with propofol and remifentanil, during which the entropy or bispectral scores were controlled between 40−60 and the hemodynamics were controlled to a maximum of 20% change from baseline values; optimal neuromuscular paralysis was maintained with rocuronium under acceleromyography monitoring. Consistent hypotension was managed with intermittent bolus dosing of either phenylephrine 100 µg or ephedrine 10 mg. Bradycardia below 50 beats/min was managed with intermittent bolus dosing of atropine 0.5 mg. Intraoperative hypothermia was prevented through the application of an air-forced blanket warmer. Incisions were made at the infraumbilical, subxyphoid, and right midclavicular subcostal regions to create access for laparoscopic cholecystectomy. At the end of surgery, patients did not receive any wound anesthetic infiltration with local anesthetics or any regional analgesia. The patients were transferred to the RR after complete reversal of the rocuronium-induced neuromuscular paralysis and when they were fully awake.

### 2.5. Interventions 

Ten minutes before the end of surgery, RP started the PCA device according to the group allocation after administration of a loading dose (0.1 mL/kg) from the PCA device along with ramosetron (0.3 mg). 

In group FN, the total PCA volume used was 120 mL, which was comprised of normal saline, fentanyl (600 μg), nefopam (120 mg), and ramosetron (1.2 mg). In group N240, the total PCA volume used was 120 mL, which was comprised of normal saline, nefopam (240 mg), and ramosetron (1.2 mg). The PCA devices were set to administer a bolus of 2 mL (fentanyl 10 μg and nefopam 2 mg, or nefopam 4 mg), with a lockout interval of 15 min and a background infusion rate of 2 mL/h (fentanyl 10 μg/h and nefopam 2 mg/h, or nefopam 4 mg/h). All drug doses were based on the ideal body weight of patients. The PCA devices were locked using a password to ensure the safe injection of drugs and to prevent changes in device settings.

When patients experienced pain of >4 points on the NRS or >40 points on the VAS, nurses or patients were allowed to push the button for administration of a bolus dose. When patients required additional rescue analgesics within the lockout interval, we permitted the intravenous injection of opioids, NSAIDs, or tramadol as a rescue analgesic to treat pain of >4 points on NRS. These rescue analgesics were selected by surgeons. When there was no consistency in the degree of pain complaints between the NRS and VAS in the RR and the ward, the patient’s postoperative pain was reevaluated using the FPRS; nurses administered a bolus dose based on an FPRS score of >4 points if it did not match the NRS score. We treated PONV of >4 points on the NRS with an intravenous injection of metoclopramide (10 mg). Our research staff decided whether to stop the PCA device based on the severity of signs and symptoms, and we excluded such cases from the final statistical analysis. 

### 2.6. Outcomes

The primary outcome of this study was the NRS at the eighth postoperative hour. We recorded pain intensity using the NRS, VAS, and FPRS, as well as the need for additional rescue analgesics and antiemetics 30 min after admission to the RR, followed by 8 and 24 h postoperatively. We downloaded data from the PCA device (cumulative infused PCA volume, per-interval bolus demand count, per-interval bolus infused count) using its built-in Wi-Fi system in 2-h intervals until 24 h postoperatively. We recorded data on demographics (age, sex, height, weight, and ASA physical status) and perioperative adverse effects.

### 2.7. Sample Size

To estimate the sample size needed for evaluation of the primary outcome, we used the online sample size calculator for non-inferiority testing (http://powerandsamplesize.com/Calculators/Compare-2-Means/2-Sample-Non-Inferiority-or-Superiority, accessed on 17 May 2019). Assuming that a difference of less than 1 point in the NRS was of no clinical importance, we selected the non-inferiority margin (δ) 1.0. Based on a previous study, the mean NRS scores were 4.20 and 4.07 12 h postoperatively in groups using nefopam monotherapy and nefopam-fentanyl polytherapy (standard deviation: 1.45), respectively [15]. The study required 70 patients in total; we thus enrolled 78 patients, allowing for a dropout rate of approximately 10%. 

### 2.8. Analysis

The non-inferiority of nefopam to fentanyl was analyzed using 95% confidence intervals (CI) for the difference in VAS. Non-inferiority was confirmed when the upper 95% CI was less than the non-inferiority margin (δ = 1.0). IBM SPSS Statistics for Windows, ver. 26.0 (IBM Corp., Armonk, NY, USA), was used for all statistical analyses. All data were analyzed as if their probability distributions were normal based on the central limit theorem, and are presented as means (95% confidence intervals [CI]), means ± standard deviation (SD), numbers of patients (*n*), or numbers (percentage) of patients (*n* [%]). We analyzed continuous variables using the Student *t*-test and nominal variables with the χ2 or Fisher’s exact test. For analysis of time-interval data that passed Mauchly’s sphericity test, we used repeated measures analysis of variance; for data that did not pass Mauchly’s sphericity test, we used Wilk’s lambda multivariate analysis of variance. To compare two groups in a given time interval, the Student *t*-test was used. *p* values < 0.05 were considered statistically significant.

## 3. Results

### 3.1. Demographic Data

There were no important harms or unintended effects in either group in this study. Seventy-eight patients were finally enrolled without any drop-outs (Table 1, Figure 1). No statistically significant differences were observed in demographic data (Table 1). 

### 3.2. Non-Inferiority Test for NRS Scores

The NRS of group N240 was not significantly inferior to that of group NF throughout the postoperative period, since the upper and lower limit of the 95% CI were within the non-inferiority margin (1.0; Table 2). 

### 3.3. NRS, VAS, and FPRS Scores

The NRS scores were not significantly different between the groups throughout the postoperative period (*p* = 0.539), and they decreased with time in both groups (*p* < 0.001; Table 2 and Figure 2a). The NRS scores of group N240 (2.41) were non-significantly different to group NF (2.31) at 8 h postoperatively (*p* = 0.605; Table 2 and Figure 2a). 

The VAS scores were not significantly different between the groups throughout the postoperative period (*p* = 0.180), and they decreased with time in both groups (*p* < 0.001, Figure 2b). The VAS scores of group N240 (23.59) were non-significantly different to group NF (23.85) at 8 h postoperatively (*p* = 0.897; Figure 2b). 

The FPRS scores were not significantly different between the groups throughout the postoperative period (*p* = 0.136), and they decreased with time in both groups (*p* < 0.001; Figure 2c). The FPRS scores of group N240 (2.36) were non-significantly different to group NF (2.44) at 8 h postoperatively (*p* = 0.709; Figure 2c).

### 3.4. Bolus Demand Count and Bolus Infused Count

The bolus demand count and bolus infused count were not significantly different between groups throughout the postoperative period (*p* = 0.270 and *p* = 0.871, respectively; Figure 3a,b), and decreased with time in both groups (*p* < 0.001 and *p* = 0.005, respectively; Figure 3a,b).

### 3.5. Cumulative Infused PCA Volume

The cumulative infused PCA volume was not significantly different between groups throughout the postoperative period (*p* = 0.495; Figure 4). 

### 3.6. Rescue Drugs and Adverse Effects

The specific postoperative rescue analgesics used were tramadol (m/c), nefopam, diclofenac, and pethidine. The proportion of patients requiring rescue analgesics and antiemetics were not significantly different between the groups throughout the recovery period (*p* = 0.225 and *p* = 0.481, respectively; Table 3). Group N240 required higher rescue analgesics than group NF (*p* ≥ 0.455; Table 3), while group NF required higher rescue antiemetics than group N240 (*p* ≥ 0.263; Table 3).

The specific postoperative adverse effects reported were PONV, hypertension, dizziness, tachycardia, and respiratory depression, which were not significantly different between the groups throughout the recovery period (*p* ≥ 0.494; Table 4). The proportion of postoperative adverse effects was not significantly different between the groups throughout the recovery period (*p* = 1.000; Table 4).

## 4. Discussion

This prospective, double-blind, randomized controlled study revealed that PCA with nefopam alone showed non-inferiority in analgesic efficacy after laparoscopic cholecystectomy; it was as effective as nefopam-fentanyl combination PCA and without significant adverse effects. 

### 4.1. Non-inferiority Test for NRS Scores

Intravenous PCA using nefopam alone (2 mg/h) was not inferior to opioid-based PCA (morphine 0.6 mg/h and ketorolac 1.8 mg/h [equivalent doses: fentanyl 6 μg/h and nefopam 1.2 mg/h]) at 12 h postoperatively [mean difference (95% CI): −0.30 (−1.25 to 0.65), non-inferiority margin: 1.5]; this produced a mean VAS of 3.3 and a lower incidence of nausea in patients undergoing laparoscopic gynecologic surgeries [13,17,18]. In this study, we also found that the postoperative analgesic effect showed non-inferiority with the non-inferiority margin (1.0) between groups using nefopam alone (4 mg/h) and combined nefopam-fentanyl (fentanyl 10 μg/h and nefopam 2 mg/h). 

### 4.2. Postoperative Pain Intensities

Two-day PCA using fentanyl alone (16 μg/kg) has been shown to be effective in controlling postoperative pain as much as 3.1 in VAS scores at 6 h postoperatively in patients undergoing laparoscopic cholecystectomy, but a high incidence of PONV (overall: 90%, very severe: 60%) was a major concern [17]. To minimize opioid-related adverse effects, we have currently adopted PCA regimens using the combination of an opioid and a non-opioid analgesic [11,12], but it was not free from opioid- and NSAID-related adverse effects, despite their reduced incidences [18]. Son et al. [18] revealed that the use of nefopam 120 mg (1 mg/h) as an adjuvant PCA analgesic with fentanyl 600 μg (5 μg/h) was effective in maintaining NRS scores of 3 to 4 at 6 h postoperatively, but resulted in a high overall incidence of PONV (59%) during the first 48 postoperative hours. Some authors have tried to evaluate the analgesic effect of intravenous nefopam alone for postoperative analgesia [13,16,18]. Postoperative analgesia with an intravenous PCA using nefopam alone was as effective as using a combination of an opioid and a non-opioid analgesic [13] and has shown non-significant differences in postoperative pain intensity compared to the use of fentanyl [16,19]. 

In this study, we set equal doses of fentanyl equivalents in each group using the following guidance: fentanyl 100 μg = morphine 10 mg = nefopam 20 mg [19]. We used nefopam 240 mg alone (4 mg/h) and nefopam 120 mg (2 mg/h) with fentanyl 600 μg (10 μg/h). If we converted nefopam to fentanyl-equivalent doses, we used a total of 1200 μg fentanyl for two-day PCA, which was about 17 μg/kg (recalculated with mean body weight) in both groups of this study, similar to that reported in Zheng et al.’s study [17]. We also found that PCA using nefopam alone was as effective in reducing postoperative pain as nefopam combined with fentanyl, which is supported by Kim et al.’s study using similar doses of nefopam and fentanyl [15]; they revealed that PCA using nefopam alone (4 mg/h) demonstrated a similar analgesic effect, bolus dose requirements, and total infused PCA volume compared to PCA using fentanyl alone (20 μg/h) or a combination of fentanyl (10 μg/h) and nefopam (2 mg/h) [15]. 

### 4.3. Nefopam-related Adverse Effects

Nefopam is associated with adverse effects such as nausea, drowsiness, light-headache, dizziness, dry mouth, and sweating, which are mostly non-significant and favorable at an appropriate dose [13,20,21,22]. In addition, nefopam is not known to cause sedation, respiratory depression, platelet dysfunction, or renal impairment [20]. However, we should pay attention to the risk of confusion and tachycardia as potential life-threatening adverse effects related to nefopam overdose [15,17,22]. Therefore, intravenous nefopam is recommended as a slow injection of single dose (20 mg) or a continuous infusion (60–120 mg/d) [21]. For this reason, we used the recommended maximum infusion rate (120 mg/d) in PCA with nefopam alone. Within recommended dosing, the incidence of cardiovascular adverse events was not significantly different between groups using PCA, with and without nefopam [14,15,17], and the incidence of nausea was not different between groups using nefopam alone and placebo [23]. We also found that the overall incidence of adverse effects was equal for groups using nefopam alone and nefopam-fentanyl (15.4%), and the most common adverse effect was nausea, with 10.3% and 15.4% in groups using nefopam alone and combined nefopam-fentanyl, respectively. Nefopam-related cardiovascular effects, such as hypertension and tachycardia, were more common in patients receiving nefopam alone, but there were no significant differences. Respiratory depression occurred in one patient in the group receiving PCA using nefopam alone, but it was related to the pethidine used as a postoperative rescue analgesic rather than the nefopam used for PCA. 

### 4.4. Opioid-Related Adverse Effects

Even though the analgesic mechanisms of nefopam are not clearly understood, the postoperative analgesic effect is known to result from its role as a serotonin–norepinephrine–dopamine reuptake inhibitor and is not related to direct action on opioid receptors or the induction of an anti-inflammatory effect as observed with NSAIDs [21,24]. Some authors have reported that a nefopam-fentanyl combination did not reduce the incidence of PONV, although this combination could be useful to decrease postoperative opioid consumption [18]. However, other authors have suggested that serotonin reuptake inhibition by nefopam could contribute to enhanced reduction in the incidence of postoperative nausea caused by opioids because serotonin is related to nausea and vomiting [13,22]. The incidence of nausea has been shown to be lower in PCA groups using nefopam alone than opioid-nefopam combinations [13], in PCA groups using nefopam alone than opioids alone [16], and in PCA groups using an opioid-nefopam combination than opioids alone [25,26]. Furthermore, premixed or bolus-injected serotonin reuptake inhibitors could contribute to reduced nausea occurring due to the administration of opioids, as well as nefopam [13]. In this study, we also premedicated ramosetron ten minutes before the end of surgery and continuously infused it via PCA devices during the postoperative period. The low incidence of opioid-related PONV in our study was influenced by the perioperative use of ramosetron. 

### 4.5. Opioid-Sparing Effect of Nefopam

PCA using a combination of nefopam (4 mg of bolus dosing) and fentanyl (10 μg of bolus dosing) showed an overall fentanyl-sparing effect of 54.5% in patients who underwent laparoscopic hysterectomy [27]. Nefopam (2.4 mg of bolus dosing), as an adjuvant PCA in addition to fentanyl (25 μg of bolus dosing), had an opioid-sparing effect of almost 34% in patients who underwent laparotomy [20]. The continuous infusion of nefopam alone (3.2 mg/hr) reduced the total fentanyl consumption by 19.3% over 48 h postoperatively [25]. In this study, we did not find a definitive fentanyl-sparing effect of nefopam, because of the non-significant differences in the total infused PCA volumes and requirement of rescue analgesics. However, since this study was not designed to specifically identify the opioid-sparing effects of nefopam, the interpretation of whether it has opioid-sparing effects should be limited to the present study results.

### 4.6. Limitations of this Study

The present study has some limitations. First, we used a fixed dose of nefopam and fentanyl for each patient, regardless of their weight. Second, the efficacy of PAC using nefopam alone is not guaranteed in surgeries with varying levels of postoperative pain intensity, because this study was conducted in laparoscopic surgery. Third, we did not assess the risk factors for PONV (female, non-smoking, motion sickness, PONV history, and postoperative opioid use) with the Apfel score, even though laparoscopic surgery is a risk factor for PONV [28]. In addition, the small sample size and the study design for a non-inferiority trial might have influenced the analysis of this study, and is a potential limitation, although there were no statistically significant differences in the incidence of bolus dose requirement and rescue analgesic administration. Therefore, these results warrant cautious interpretation, and further studies evaluating the efficacy of PCA using nefopam alone still need to be performed in different types of surgeries with a larger sample size.

## 5. Conclusions

Intravenous PCA using nefopam alone has a non-inferior and effective analgesic efficacy and produces a lower incidence of opioid- and nefopam-related adverse effects compared to a combination of fentanyl and nefopam after laparoscopic cholecystectomy.

## Figures and Tables

**Figure 1 medicina-57-00316-f001:**
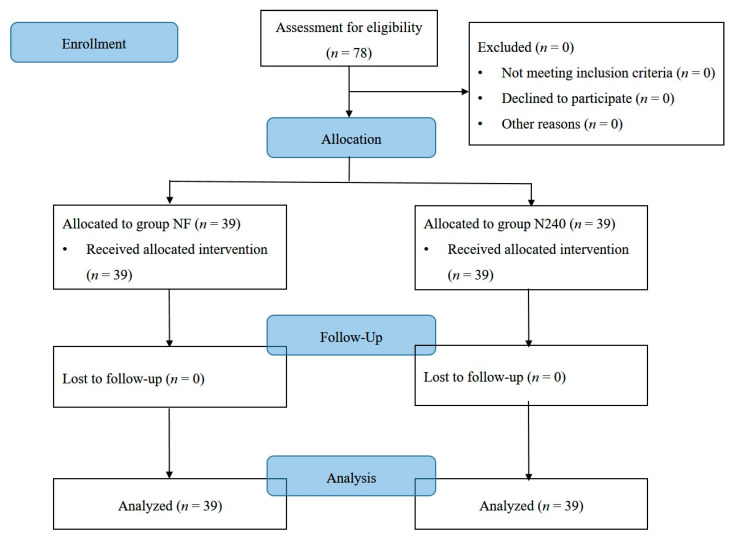
CONSORT diagram for patient recruitment. Group NF, group receiving nefopam-fentanyl polytherapy; Group N240, group receiving nefopam monotherapy.

**Figure 2 medicina-57-00316-f002:**
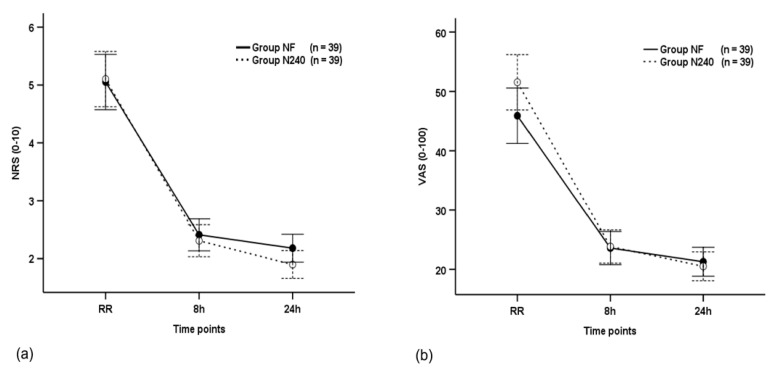
Time-sequential changes in numeric rating scale (NRS) scores (**a**), visual analogue scale (VAS) scores (**b**), and Faces Pain Ratings Scale (FPRS) scores (**c**). Data points and error bars represent means and 95% confidence intervals, respectively. Group NF, group receiving nefopam-fentanyl polytherapy; Group N240, group receiving nefopam monotherapy; RR, 30 min after admission to the recovery room.

**Figure 3 medicina-57-00316-f003:**
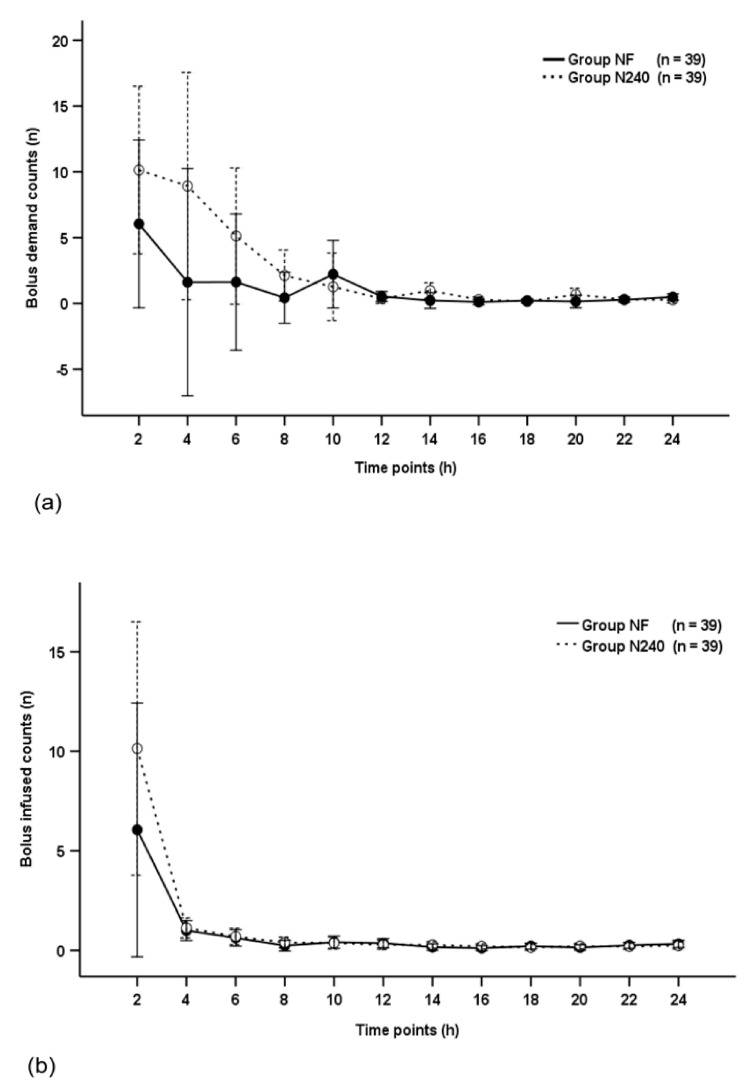
Time-sequential changes of bolus demand counts (**a**) and bolus infused counts (**b**). Data points and error bars represent means and 95% confidence intervals, respectively. Group NF, group receiving nefopam-fentanyl polytherapy; Group N240, group receiving nefopam monotherapy.

**Figure 4 medicina-57-00316-f004:**
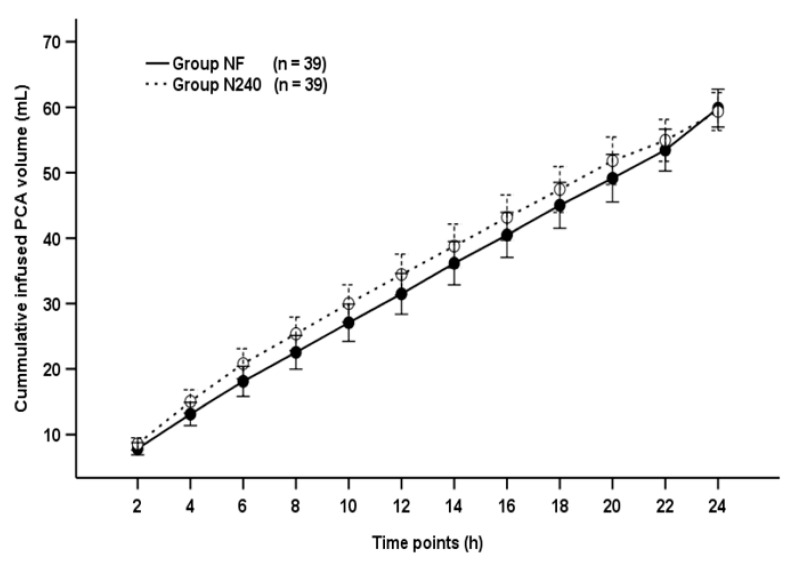
Time-sequential changes of cumulative infused PCA volumes. Data points and error bars represent means and 95% confidence intervals, respectively. Group NF, group receiving nefopam-fentanyl polytherapy; Group N240, group receiving nefopam monotherapy.

**Table 1 medicina-57-00316-t001:** Demographic data and intraoperative variables.

Variables	Group NF (*n* = 39)	Group N240 (*n* = 39)	*p* Value
Age (yr)	50.5 ± 14.1	50.4 ± 11.3	0.972
Sex (M/F)	23/16	26/13	0.482
Height (cm)	164.9 ± 10.1	166.5 ± 8.2	0.440
Weight (kg)	69.8 ± 13.4	70.7 ± 11.8	0.754
ASA-PS (I/II/III)	11/25/3	10/27/2	0.850

Values are expressed as the means ± standard deviation or number of patients. Group NF, group receiving nefopam-fentanyl polytherapy; Group N240, group receiving nefopam monotherapy; ASA-PS, American Society of Anesthesiologists physical status.

**Table 2 medicina-57-00316-t002:** Test for non-inferiority between nefopam and nefopam-fentanyl in patient-controlled analgesia following laparoscopic cholecystectomy.

Variables		Group NF (*n* = 39)	Group N240 (*n* = 39)	*p* Value	Difference between Groups (95% CI)
NRS					
Time points	RR	5.05 (4.59−5.52)	5.10 (4.59−5.61)	0.881	−0.05 (−0.73 to 0.63)
	8 h	2.41 (2.07−2.75)	2.31 (2.10−2.52)	0.605	0.10 (−0.29 to 0.50)
	24 h	2.18 (1.90−2.46)	1.90 (1.69−2.11)	0.103	0.28 (−0.06 to 0.62)

Values are expressed as the means (95% confidence intervals [CI]). Group NF, group receiving nefopam-fentanyl polytherapy; Group N240, group receiving nefopam monotherapy; NRS, numeric rating scale; RR, 30 min after admission to the recovery room.

**Table 3 medicina-57-00316-t003:** Number of patients receiving rescue analgesics and antiemetics.

			Time		
Variables	Groups	RR	8 h	24 h	Total
Analgesics	Group NF (*n* = 39)	3 (7.7)	5 (12.8)	3 (7.7)	10 (25.6)
	Group N240 (*n* = 39)	6 (15.4)	6 (15.4)	5 (12.8)	15 (38.5)
	*p* value	0.481	0.745	0.455	0.225
Antiemetics *	Group NF (*n* = 39)	0 (0)	6 (15.4)	0 (0)	6 (15.4)
	Group N240 (*n* = 39)	0 (0)	2 (5.1)	0 (0)	2 (5.1)
	*p* value	-	0.263	-	0.263

Values are expressed as the number (percentage) of patients [*n* (%)]. Group NF, group receiving nefopam-fentanyl polytherapy; Group N240, group receiving nefopam monotherapy; RR, 30 min after admission to the recovery room. *: number (percentage) of patients who experienced PONV with NRS score >4.

**Table 4 medicina-57-00316-t004:** Adverse effects.

	Group NF (*n* = 39)	Group N240 (*n* = 39)	*p* Value
PONV *	6 (15.4)	4 (10.3)	0.737
HTN	0 (0)	1 (2.6)	1.000
Dizziness	2 (5.1)	0 (0)	0.494
Tachycardia	0 (0)	1 (2.6)	1.000
Respiratory depression	0 (0)	1 (2.6)	1.000
Total	6 (15.4)	6 (15.4)	1.000

Values are expressed as the number (percentage) of patients [*n* (%)]. Group NF, group receiving nefopam-fentanyl polytherapy; Group N240, group receiving nefopam monotherapy; HTN, hypertension; PONV, postoperative nausea and vomiting. *: number (percentage) of patients who experienced PONV with NRS score >0.

## Data Availability

The data presented in this study are available on request from the corresponding author, through institutional review board, and reviewers. The data are not publicly available due to restrictions of obtaining approval from the IRB for the disclosure of data. If anyone requires our data of this study, please do not hesitate to contact the corresponding author.

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
