# Peer review of "Effect of Nefopam-Based Patient-Controlled Analgesia with and without Fentanyl on Postoperative Pain Intensity in Patients Following Laparoscopic Cholecystectomy: A Prospective, Randomized, Controlled, Double-Blind Non-Inferiority Trial"

_medicina, 2021, doi:10.3390/medicina57040316_

Round 1
Reviewer 1 Report
This study was so interesting in terms of new analgesic method after laparoscopic cholecystectomy. I have some comments.
- Why did you think to use this nefopam? What was your background? This nefopam had opioid-sparing effect?
- Regional anesthesia was standard for this type’s surgery. Why didn’t they use regional anesthesia?
- Please describe the surgical wound.
- Did they definite protocol regarding other analgesics? Who decided other analgesics?
- Who collected NRS information?
Author Response
Here is a point-by-point response to the reviewers’ comments and concerns.
Response to Reviewer 1 Comments
- We used the line number in the manuscript, which maintained the "Track Changes" function in Microsoft Word.
Point 1: Why did you think to use this nefopam? What was your background? This nefopam had opioid-sparing effect?
Response 1: Thank you for pointing this out. We had used NSAIDs as an adjuvant analgesic considering the known opioid-sparing effect to reduce the amount of opioids. Nefopam is a centrally acting non-opioid, non-steroidal analgesic that has been used as an alternative to opioids for analgesia in patients with moderate to severe pain. Nefopam as an adjuvant analgesic for fentanyl-based PCA has been shown to provide similar postoperative analgesia to ketorolac, a common NSAID used as an adjuvant analgesic with fentanyl-based PCA. So, recently, we have mainly used nefopam instead of NSAIDs, if it does not correspond to contraindication, according to our PCA protocol.
There are several articles reporting that nefopam had opioid-sparing effects (19.3% −54.5%). In this study, we did not found that there was definitive fentanyl-sparing effects of nefopam because of non-significant difference of total infused PCA volumes and requirement of rescue analgesics. However, since this study was not designed to identify opioid-sparing effects of nefopam, it is limited to interpreting whether it has opioid-sparing effects based on the results of this study.
So, we further described the opioid-sparing effect of nefopam in a new paragraph of discussion (in lines 359-370):
“4.5. Opioid-sparing effect of nefopam
PCA using a combination of nefopam (4 mg of bolus dosing) and fentanyl (10 μg of bolus dosing) showed an overall fentanyl-sparing effect of 54.5% in patients who underwent laparoscopic hysterectomy [28]. Nefopam (2.4 mg of bolus dosing), as an adjuvant PCA in addition to fentanyl (25 μg of bolus dosing), had an opioid-sparing effect of almost 34% in patients who underwent laparotomy [20]. The continuous infusion of nefopam alone (3.2 mg/hr) reduced the total fentanyl consumption by 19.3% over 48 hours postoperatively [25]. In this study, we did not find a definitive fentanyl-sparing effect of nefopam because of the non-significant differences in the total infused PCA volumes and requirement of rescue analgesics. However, since this study was not designed to specifically identify the opioid-sparing effects of nefopam, the interpretation of whether it has opioid-sparing effects should be limited to the present study results.”
Point 2: Regional anesthesia was standard for this type’s surgery. Why didn’t they use regional anesthesia?
Response 2: Thank you for pointing this out. Regional anesthesia such as spinal and epidural anesthesia was not standard protocol for laparoscopic cholecystectomy in our hospital except of patients with special circumstances. If regional anesthesia means regional block, we sometimes performed the regional abdominal wall block or epidural block (continuous or intermittent) for post-laparoscopic analgesia, but it also was not standard protocol. Therefore, we are using IV PCA mainly for postoperative analgesia as standard protocol for postoperative analgesia. Please consider this point.
Point 3: Please describe the surgical wound.
Response 3: Thank you for pointing this out. In our hospital, general surgeon performed laparoscopic cholecystectomy with three-port technique via open wound with infraumbilical, subxyphoid, and right midclavicular subcostal incision. We further described the open wounds for laparoscopic cholecystectomy as follows (in lines 136-137):
“Intraoperative hypothermia was prevented through the application of an air-forced blanket warmer. Incisions were made at the infraumbilical, subxyphoid, and right midclavicular subcostal regions to create access for laparoscopic cholecystectomy. At the end of surgery, patients did not receive any wound anesthetic infiltration with local anesthetics or any regional analgesia.”
Point 4: Did they definite protocol regarding other analgesics? Who decided other analgesics?
Response 4: Thank you for pointing this out. The surgical part of our hospital has its own protocol for postoperative management, which includes additional administration of painkillers if receiving PCA. However, for patients participating in clinical trials, we (researchers) trained nurses and surgeons on the condition that additional rescue analgesics could be used for those patients, and we allowed surgeons to administer the rescue analgesics decided by themselves. We described it as follows (in line 159):
“When patients experienced pain of >4 points on the NRS or >40 points on the VAS, nurses or patients were allowed to push the button for administration of a bolus dose. When patients required additional rescue analgesics within the lockout interval, we permitted the intravenous injection of opioids, NSAIDs, or tramadol as a rescue analgesic to treat pain of >4 points on NRS. These rescue analgesics were selected by surgeons. When there was no consistency in the degree of pain complaints between the NRS and VAS in the RR and the ward, the patient’s postoperative pain was reevaluated using the FPRS; nurses administered a bolus dose based on an FPRS score of >4 points if it did not match the NRS score. We treated PONV of >4 points on the NRS with an intravenous injection of metoclopramide (10 mg).”
Point 5: Who collected NRS information?
Response 5: Thank you for pointing this out. NRS was recorded by nurses trained by the hospital to assess pain intensity, and researcher who managed the anesthesia (RA) was finally collected NRS information from the nursing medical records. We already described these information as follows (in lines 112-125). Please take this into consideration.
“The researcher who managed the anesthesia (RA) was responsible for obtaining informed consent from participants, as well as gathering and recording data from the participants and PCA devices. The researcher who managed the PCAs (RP) was responsible for assigning the correct drugs to each PCA device according to the randomization scheme. For blinding, RP recorded the drug assignment on anesthetic charts after the anesthesia was completely finished, and RA finally collated the data from patient medical records as well as data generated through the trial for at least 24 hours postoperatively. Neither RA nor RP participated in the statistical analysis.
The nurses in the recovery room (RR) or ward recorded data on postoperative pain and postoperative nausea and vomiting (PONV) using the NRS; these nurses were not part of the investigating team and were trained by the hospital to assess pain intensity and PONV using the NRS, visual analogue score (VAS: 0 = no pain, 10 = worst pain imaginable), or Woong-Baker Faces Pain Ratings Scale (FPRS: 0=no pain, 10=most severe pain).”

Reviewer 2 Report
In their manuscript, entitled „Effect of nefopam-based patient-controlled analgesia with and without fentanyl on postoperative pain intensity in patients following laparoscopic cholecystectomy: A prospective, randomized, controlled, double-blind non-inferiority trial”, the authors present results from a non-inferiority study evaluating the efficacy of nefopam, a non-opioid analgesic, in the setting of patient-controlled analgesia.
The manuscript is well-written, the methods including the statistical analysis is sound and the results are presented very well. However, I have some issues with the manuscript, which I would kindly ask the authors to address.
Major:
- In the Methods section, the authors report that “one hundred sixty were randomly assigned” to the two groups. However, all other numbers say that there were 79 patients. Please clarify!
- The non-inferiority trial design with the well-presented statistical methods allows the authors to rightfully state that – in this study – nefopam alone did not show inferiority compared to nefopam with fentanyl. However, looking at the results I still have some issues with the interpretation:
- The mean bolus demand counts during the first 6 hours after surgery (Figure 3) were a lot higher (6 vs 10 at 2 hrs, 2 vs 9 at 4hrs, etc.) in the N240 group, which is not surprising due to the fact that the only opioid used here has been remifentanil, a very short acting substance. However, due to the high variance and the rather small n of 39, of course no statistically significant differences could be detected, although the clinical impact might still be there.
- The same was true for the application of rescue analgesics in the RR: The number of patients receiving rescue analgesics was actually doubled in the N240 group (3 vs 6 patients)! Again, no statistically significant results, which is correct.
- Interestingly, the authors have stated that PONV has been one of the major concerns against the use of an opioid-based PCA regimen, which is one reason they initiated this study. However, the rate of PONV was not significantly different at all (table 4).
- Additionally, there were 4 patients with PONV in the N240 group (table 4), but only 2 received an antiemetic (table 3). Please clarify!
Therefore, in my opinion, the interpretation of the results should be made with a lot more caution, due to the fact that the study was rather small in size and designed as a non-inferiority trial. This should also be reflected in the discussion and I would kindly ask the authors to revise it accordingly.
Additional comment:
Methods, L126: Just a comment on the side: Do you really think that intramuscular midazolam is an adequate and patient-comfort-friendly way of premedication? I personally think that there are other routes of application available, which are way more appropriate these days.
Minor:
The last sentence of the introduction (LL 80-83) should be omitted. This is already part of the discussion.
Author Response
Response to Reviewer 2 Comments
- We used the line number in the manuscript, which maintained the "Track Changes" function in Microsoft Word.
Major revisions:
Point 1: In the Methods section, the authors report that “one hundred sixty were randomly assigned” to the two groups. However, all other numbers say that there were 79 patients. Please clarify!
Response 1: Thank you for pointing this out. We agree with this comment. We found that the number of patients was mistyped to ‘One hundred sixty’ instead of ‘Seventy eight’ in the method section. So, we revised the number of patients exactly to ‘Seventy eight’ (2.3. Randomization and Masking) as follows (in line 106).
“2.3. Randomization and Masking
Seventy-eight One hundred sixty patients were randomly assigned to two groups that received PCA with either a combination of fentanyl 600 μg and nefopam 120 mg (group NF, n = 39) or nefopam 240 mg alone (group N240, n = 39). Randomization was performed using a computer-generated table of random numbers with 1:1 allocation ratio. This randomization was performed using an online website (https://www.graphpad.com/quickcalcs/).”
Point 2: The non-inferiority trial design with the well-presented statistical methods allows the authors to rightfully state that – in this study – nefopam alone did not show inferiority compared to nefopam with fentanyl. However, looking at the results I still have some issues with the interpretation:
- The mean bolus demand counts during the first 6 hours after surgery (Figure 3) were a lot higher (6 vs 10 at 2 hrs, 2 vs 9 at 4hrs, etc.) in the N240 group, which is not surprising due to the fact that the only opioid used here has been remifentanil, a very short acting substance. However, due to the high variance and the rather small n of 39, of course no statistically significant differences could be detected, although the clinical impact might still be there.
- The same was true for the application of rescue analgesics in the RR: The number of patients receiving rescue analgesics was actually doubled in the N240 group (3 vs 6 patients)! Again, no statistically significant results, which is correct.
- Interestingly, the authors have stated that PONV has been one of the major concerns against the use of an opioid-based PCA regimen, which is one reason they initiated this study. However, the rate of PONV was not significantly different at all (table 4).
- Additionally, there were 4 patients with PONV in the N240 group (table 4), but only 2 received an antiemetic (table 3). Please clarify!
Therefore, in my opinion, the interpretation of the results should be made with a lot more caution, due to the fact that the study was rather small in size and designed as a non-inferiority trial. This should also be reflected in the discussion and I would kindly ask the authors to revise it accordingly.
Response 2: Thank you for pointing this out. We agree with this comment. So, we included it into the limitation as follows (in lines 378-384):
“In addition, the small sample size and the study design for a non-inferiority trial might have influenced the analysis of this study and is a potential limitation, although there were no statistically significant differences in the incidence of bolus dose requirement and rescue analgesic administration. Therefore, these results warrant cautious interpretation, and studies evaluating the efficacy of PCA using nefopam alone still need to be performed in different types of surgeries and with larger sample size.”
Regarding the reviewer's comment on PONV: It is well known that opioid-based PCA is associated with the high incidence of PONV. We also expected the high incidence of PONV in patients receiving opioid-based PCA. However, in this study, the incidence of PONV was about 15%, showing no statistical difference between nefopam alone (10.3%). So, we described that the premedication of ramosetron (antiemetics) before the end of surgery and continuous infusion of ramosetron (antiemetics) along with opioid during PCA might have influenced low incidence of PONV in the section of [4.4. Opioid-related Adverse Effects] (in lines 353-358).
“Furthermore, premixed or bolus-injected serotonin reuptake inhibitors could contribute to reduced nausea occurring due to the administration of opioids as well as nefopam [13]. In this study, we also premedicated ramosetron ten minutes before the end of surgery and continuously infused it via PCA devices during the postoperative period. The low incidence of opioid-related PONV in our study was influenced by the perioperative use of ramosetron.”
In addition, we treated PONV of >4 points on the NRS with an intravenous injection of antiemetics. Therefore, only 2 patients received antiemetics despite 4 patients complained PONV. We described it in Tables 3 and 4.
In table 3, “*: number (percentage) of patients who experienced PONV with NRS score >4.” In table 4, “*: number (percentage) of patients who experienced PONV with NRS score >0..”
Finally, please take into consideration that the main outcome of this study was the NRS, postoperative pain score.
Other:
Point 3: Additional comment: Methods, L126: Just a comment on the side: Do you really think that intramuscular midazolam is an adequate and patient-comfort-friendly way of premedication? I personally think that there are other routes of application available, which are way more appropriate these days.
Response 3: Thank you for pointing this out. Like the comment of reviewer, we know and agree that there are various routes of premedication to induce anxiolysis and sedation other than intramuscular injections. However, many anesthesiologists in Korea still prefer to use intramuscular injections, and our hospitals are also using these methods according to the preanesthetic premedication protocol. Please take this into consideration.
Point 4: Minor: The last sentence of the introduction (LL 80-83) should be omitted. This is already part of the discussion.
Response 4: Thank you for pointing this out. According to the reviewer's comment, we deleted the last sentence of the introduction (LL 80-83).
“The primary outcome of this study was whether the non-inferiority margin of the numeric rating scale (NRS) exceeded 1.0 at 8 hours postoperatively in the group receiving PCA using nefopam alone (group N240) compared to that using a nefopam-fentanyl combination (group NF). We found that PCA using nefopam alone has a non-inferior and effective analgesic efficacy and produces a lower incidence of postoperative ad-verse effects compared to a combination of fentanyl and nefopam after laparoscopic cholecystectomy.”
We look forward to hearing from you in due time regarding our submission and to respond to any further questions and comments you may have.
Sincerely,
[Sang Hun Kim, 17th, MAR., 2021]
